# Breast-Specific Gamma Imaging with [^99^mTc]Tc-Sestamibi: An In Vivo Analysis for Early Identification of Breast Cancer Lesions Expressing Bone Biomarkers

**DOI:** 10.3390/jcm9030747

**Published:** 2020-03-10

**Authors:** Nicoletta Urbano, Manuel Scimeca, Carmela Di Russo, Elena Bonanno, Orazio Schillaci

**Affiliations:** 1Nuclear Medicine, Policlinico “Tor Vergata”, viale Oxford, 81, 00133 Rome, Italy; n.urbano@virgilio.it (N.U.); carmela.dirusso@ptvonline.it (C.D.R.); 2Department of Biomedicine and Prevention, University of Rome “Tor Vergata”, Via Montpellier 1, 00133 Rome, Italy; manuel.scimeca@uniroma2.it; 3San Raffaele University, Via di Val Cannuta 247, 00166 Rome, Italy; 4Fondazione Umberto Veronesi (FUV), Piazza Velasca 5, 20122 Milano, Italy; 5Saint Camillus International University of Health Sciences, Via di Sant’Alessandro, 8, 00131 Rome, Italy; 6Department of Experimental Medicine, University “Tor Vergata”, Via Montpellier 1, 00133 Rome, Italy; elena.bonanno@uniroma2.it; 7Diagnostica Medica’ & ‘Villa dei Platani’, Neuromed Group, 83100 Avellino, Italy; 8IRCCS Neuromed, 86077 Pozzilli, Italy

**Keywords:** breast cancer, sestamibi, imaging, breast-specific gamma imaging, biomarkers

## Abstract

The main purpose of this pilot investigation was to evaluate the possible relationship among [^99^mTc]Tc-Sestamibi uptake, the presence of breast osteoblast-like cells, and the expression of molecules involved in bone metabolism, such as estrogen receptor, bone morphogenetic proteins-2, and PTX3. To this end, forty consecutive breast cancer patients who underwent both breast-specific gamma imaging with [^99m^Tc]Tc-Sestamibi and breast bioptic procedure were retrospectively enrolled. From each diagnostic paraffin block collected in the study, histological diagnosis, immunohistochemical investigations, and energy dispersive X-ray microanalysis were performed. Our data highlight the possible use of breast-specific gamma imaging with [^99^mTc]Tc-Sestamibi for the early detection of breast cancer lesions expressing bone biomarkers in the presence of breast osteoblast-like cells. Specifically, we show a linear association among sestamibi uptake, the presence of breast osteoblast-like cells, and the expression of estrogen receptor, bone morphogenetics proteins-2, and PTX3. Notably, we also observed an increase of [^99^mTc]Tc-Sestamibi in breast cancer lesions with magnesium-substituted hydroxyapatite. In conclusion, in this pilot study we evaluated data from the nuclear medicine unit and anatomic pathology department on breast cancer osteotropism, identifying a new possible interpretation of Breast Specific Gamma Imaging with [^99^mTc]Tc-Sestamibi analysis.

## 1. Introduction

Carcinomas of the breast represent one of the most common malignant tumors in the world with an incidence rate of 10.4% of all cancers [1]. Moreover, breast cancer is the leading cause of death in women aged between 20 and 50 years [1]. Recent statistical studies reported that more than 1.7 million new cases of cancers are expected to be diagnosed in 2019 in the United States. Among them, 271,000 will be breast cancer (268,600 women and 2670 in men), of which 42,263 will be fatal (41,760 in women and 500 in men) [2]. Breast cancer alone accounts for 30% of all new cancer diagnoses in women. It is still the second cause of death from cancer, being responsible of 15% of cancer deaths, after those for lung cancer (30%) and before those for colon rectal cancer (8%) [3]. Breast cancer metastasis, defined as the homing of breast cancer beyond the ipsilateral breast, chest wall, and regional lymph nodes, remains the main cause of death. In 6%–10% of cases, there is already metastasis at the moment of diagnosis, and almost 30% of early-stage breast cancers will give recurrence or metastasis [4]. Despite the recent diagnostic and therapeutic advance in the management of breast cancer, the prognosis in metastatic lesions remains poor. Bone, lymph nodes, brain, liver, lung, skin besides breast or chest wall, ovaries, spinal cord, eye, and heart are the most frequent sites of breast metastasis. It is known that the development of osteolytic bone metastatic lesions in breast cancer heavily affects patients’ quality of life due to the occurrence of bone fractures, bone pain, spinal cord compression, and hypercalcemia [1]. A deep understanding of the mechanisms by which distant metastasis develop and involve specific sites, mainly the bone, is of crucial importance and of clinical value. Breast cancer metastasis to the bone, as for most malignant tumors, is not a random process, but there is a predilection for tumor homing into specific organs where tumor cells nest—the so-called “organotropism of metastasis” or “non-random organ specific metastasis” [5,6,7].

In this scenario, Scimeca et al. reported a new vision about the possible involvement of breast osteoblast-like cells (BOLCs) in the carcinogenesis of lesions with a high propensity to form bone metastasis [8,9,10]. In one of these studies, the authors reported very preliminary data about the possible use of breast-specific gamma imaging (BSGI) with [^99^mTc]Tc-Sestamibi for the detection of breast cancer lesions, characterized by the presence of BOLCs [8] and thus susceptible to bone metastasis. Despite these preliminary data, however, no diagnostic evaluation is currently available for early detection of breast cancer lesions expressing bone biomarkers.

Recently, the American College of Radiology reported detailed indications for the use of dedicated breast gamma imaging, Breast Specific Gamma Imaging—(BSGI) and Molecular breast imaging (MBI), in order to study the extent of disease/preoperative staging in de novo detected breast tumors, assessment of response to the treatment with neoadjuvant chemotherapy, identification of local recurrence, assessment for primary breast lesions in patients with unidentified primary, and screening for high risk women who cannot undergo MRI and MBI as adjunct to conventional breast imaging for problem solving in unclassified cases [11,12].

Starting from all these considerations, the main aim of this pilot investigation was to evaluate the potential role of BSGI with [^99^mTc]Tc-Sestamibi in the detection of breast cancer lesions expressing bone biomarkers given the presence of BOLCs, or microcalcifications. Specifically, here, we report the correlation analysis among the uptake of ^99^mTc sestamibi, the presence of BOLCs, and the expression of their main markers, microcalcifications included, on a cohort of 40 consecutive patients who underwent both BSGI with [^99^mTc]Tc-Sestamibi and breast biopsy procedure.

## 2. Methods

The “Policlinico Tor Vergata” Ethical Committee approved this protocol with reference number 129.18. In addition, all methodologies and experimental procedures described here were achieved in agreement with the latest Helsinki Declaration.

Exclusion criteria are a second cancer and neoadjuvant hormonal or radiation therapy prior to surgery. According to these criteria, we retrospectively enrolled 40 consecutive breast cancer patients (58.36 ± 1.99 years; range of 42–65 years) who underwent both BSGI with [^99^mTc]Tc-Sestamibi and breast bioptic procedure from January 2018 to November 2018. For each patient, histological diagnosis, immunohistochemical investigations, and energy dispersive X-ray (EDX) microanalysis were performed.

### 2.1. [^99^mTc]Tc-Sestamibi–High Resolution SPECT

BSGI with [^99^mTc]Tc-Sestamibi investigations were performed as described in a previous study [13]. Briefly, a BSGI scan was performed in 10–15 min following an intravenous administration of 740 MBq Tc-99m-MIBI (Bristol-Myers Squibb Pharma, Bruxelles, Belgium) through an antecubital vein contralateral to the suspicious breast side to avoid potential false-positive uptake in the axillary lymph nodes. The patients remained seated during the procedure. Craniocaudal (CC) and mediolateral oblique (MLO) images were obtained in both breasts using a high-resolution BSGI (Millennium VG & Hawkeye; General Electric Medical Systems, Milwaukee, WI, USA).

All 40 patients had a biopsy. BSGI was performed before biopsy in 25 patients and after biopsy in 15 patients. When BSGI was performed after biopsy, the minimum interval between biopsy and imaging was 7 days in an effort to avoid the effects of post-biopsy inflammation as much as possible.

For qualitative analysis of BSGI, two investigators classified positive and negative findings. Lesions with no demonstrable uptake and those with diffuse heterogeneous or minimal patchy uptake were considered negative, whereas lesions with scattered patchy uptake, partially focal uptake, or any other focal uptake were regarded as positive. Irregular-shaped regions of interest (ROIs) were used to encase the lesions. The evaluation of the lesion to non-lesion ratio (L/N) were estimated according to the study of Tan et al. [14]. For patients who underwent BSGI with sestamibi before biopsy (*n* = 25), the BSGI-guided biopsy procedure was performed as previously described [15]. The semi-quantitative of the [^99^mTc]Tc-Sestamibi was performed.

### 2.2. Histology

Breast bioptic samples were formalin fixed and embedded in Paraffin. Serial sections were used for both haematoxylin–eosin (H&E) and immunohistochemistry staining [16].

### 2.3. Immunohistochemistry

Immunohistochemistry was used to study the presence of BOLCs in collected samples, the expression of estrogen receptor (ER), PTX3 and bone morphogenetic protein(BMP)-2, and the proliferation index by Ki67. Three-μm-thick paraffin sections were treated with EDTA citrate buffers pH 7.8 for 30 min at 95 °C to antigen retrieval reaction. Afterwards, sections were incubated with the primary antibodies diluted 1:100 for 60 min at room temperature: anti-RUNX2 mouse monoclonal antibody (clone 1D8, Novus Biologicals, Centennial, CO 80112, USA), anti-RANKL mouse monoclonal antibody (clone 12A668, AbCam, Cambridge, UK) anti-Ki67 rabbit monoclonal antibody (clone 30-9, Ventana, Tucson, USA), anti-ER rabbit monoclonal antibody (clone SP1, Novus Biologicals, Centennial, CO 80112, USA), anti-BMP-2 mouse monoclonal antibody (clone 9E10G12; Novus Biologicals, Centennial, CO 80112, USA), and anti-PTX3 rat monoclonal antibody (clone MNB1; Abcam, UK). Reactions were detected by using HRP-DAB Detection Kit (UCS Diagnostic, Italy) [17]. RUNX2 (streptavidin-texas-red) and RANKL (streptavidin-FITC) were detected simultaneously by immunofluorescence. Specifically, after incubation for 60 min at room temperature, reaction with anti-Runx2 was revealed by using FITC-conjugated anti-mouse antibody. Afterwards, sections were incubated with anti-RANKL mouse monoclonal antibody for 60 min. Reaction with anti-RANKL antibody was revealed by using Texas Red-conjugated anti-mouse antibody. Washing was performed with PBS/Tween 20 pH 7.6 (UCS Diagnostic). Immunofluorescence reactions were evaluated by using the Nikon Eclipse e-1000 microscope (1300 Walt Whitman Road Melville, NY 11747-3064 USA).

Digital scans were used to evaluate the immunohistochemical reactions (Iscan Coreo, Ventana, Tucson, AZ, USA). Specifically, digital images from ER, BMP-2, and PTX3 reactions were evaluated in a semi-quantitative approach by counting the number of positive breast cancer cells (out of a total of 500 in randomly selected regions). Ki67 was calculated in terms of the percentage of positive cancer cells. Reactions have been set up by using specific positive and negative control tissues.

### 2.4. EDX Microanalysis

All breast samples underwent EDX microanalysis, as previously described [18]. EDX spectra of calcium deposits were acquired with a transmission electron microscope (Hitachi, Schaumburg, IL, USA) and an EDX detector (Thermo Scientific, Waltham, MA, USA) at an acceleration voltage of 75 KeV and magnification of 12,000. Spectra were semi-quantitatively acquired using the standardless Cliff-Lorimer k-factor method.

### 2.5. Statistical Analysis

In order to evaluate the possible association among sestamibi uptake, the presence BOLCs, and the expression of ER, Ki67, BMP-2, and PTX3, linear regression analyses were performed.

## 3. Results

### 3.1. ^99m^Tc-Sestamibi–High Resolution SPECT Analysis

BSGI with [^99^mTc]Tc-Sestamibi showed sestamibi uptake in all 40 patients (L/N max 5,30; min 1,41) (Figure 1A). No significant differences concerning L/N Ratio were observed among breast cancer histotypes (data not shown).

### 3.2. Histology

Breast biopsies were classified according to the Nottingham Histological system [19]. In particular, we found 10/40 G1 infiltrating carcinomas, 19/40 G2 infiltrating carcinomas, and 11/40 G3 infiltrating carcinomas.

### 3.3. Sestamibi Uptake vs. BOLCs

As shown in Figure 1, we observed a positive and significant correlation between the BOLC number in collected breast cancer tissues and the uptake of [^99^mTc]Tc-Sestamibi (r^2^ = 0.5056; *p* < 0.0001) (Figure 1B–D). On this note, breast lesions showing a higher number of BOLCs > 200/500 were characterized by a very high value of L/M. A significant correlation between [^99^mTc]Tc-Sestamibi uptake and cancer proliferation index (percentage of Ki67 positive cancer cells) was also observed (r^2^ 0.3162; *p* = 0.0002) (Figure 1F,G).

### 3.4. Sestamibi Uptake vs. BOLCs’ Main Markers

To test the assumption that BSGI with [^99^mTc]Tc-Sestamibi is capable to identify breast tumors with a high number of BOLCs (Figure 2A), we investigated the possible association between the L/N values and the expression of the main biomarkers of the BOLCs. Notably, the uptake of [^99^mTc]Tc-Sestamibi displayed a very significant association with the number of ER positive cells (r^2^ = 0.1527; *p* < 0.0208) (Figure 2B). Lastly, a significant association between both BMP-2 (Figure 2C) and PTX3 (Figure 2D) expression and [^99^mTc]Tc-Sestamibi uptake was noted (BMP-2 r^2^ = 0.4953; *p* < 0.0001; PTX3 r^2^ = 0.2975; *p* = 0.0003).

### 3.5. EDX Microanalysis

Calcifications were classified as HA and MgHAp by EDX microanalysis. Specifically, we detected 1/15 CO, 5/15 HA, and 9/15 MgHAp.

### 3.6. Sestamibi Uptake vs. Microcalcifications

As concerns the study of the association between the presence of breast microcalcifications and the [^99^mTc]Tc-Sestamibi uptake, we observed no significant association (Micro+ vs. Micro− *p* = 0.0.5025) (Figure 3A). Notably, we noted a significant difference in [^99^mTc]Tc-Sestamibi uptake when considering the elemental composition of microcalcifications (Figure 3B) and HA (Figure 3C) vs. MgHAp (Figure 3D).

## 4. Discussion

The identification of cellular and molecular processes involved in the development of skeletal lesions can provide a scientific rationale for designing an in vivo analysis for the early detection of breast tumors with a high propensity to form bone metastasis due to the expression of bone biomarkers.

Considering the use of molecular imaging analysis, BSGI with [^99^mTc]Tc-Sestamibi can represent an unexpected opportunity to detect both primary and metastatic lesions characterized by the presence of BOLCs. Indeed, chemical–physical characteristics of sestamibi, as well as its pharmacokinetics, appear suitable for in vivo identification of BOLCs. Chemically, [^99^mTc]Tc-Sestamibi is a lipophilic cation member of the isonitrile family (hexakis 2-methoxyisobutyl isonitrile) that accumulates in the cytoplasm of breast cells as a result of passive diffusion across the plasma membrane [20]. Once in the cytoplasm, [^99^mTc]Tc-Sestamibi can accumulate inside the mitochondria due to their negative membrane potential [21]. The positive charge on sestamibi may drive this molecule into the mitochondria during cell metabolic activities that increase the negative plasma membrane potential. On this note, the in vitro experiments displayed an increase of the high-transmembrane-potential mitochondria in osteoblast or osteoblast-like cells induced by molecules capable to induce the formation of bone matrix, such as estrogen [22]. In light of this, it is possible to speculate an increase of the uptake of sestamibi in breast cancers with a high number of BOLCs due to the increase of the high-transmembrane-potential mitochondria during the formation of microcalcifications made of HA that are generally produced by the BOLCs themselves. In agreement with this evidence, ultrastructural observations demonstrated the presence of several mitochondria, both in osteoblast-like cells and real osteoblasts during calcification production [10,23,24]. Therefore, BSGI with [^99^mTc]Tc-Sestamibi could be used to stratify breast cancer patients based on the risk of developing bone metastases given the presence of a high number of BOLCs.

The main aim of this pilot investigation was to evaluate the potential role of BSGI with [^99^mTc]Tc-Sestamibi in the detection of breast cancer lesions expressing bone biomarkers given the presence of BOLCs, or microcalcifications. To this end, we enrolled 40 consecutive patients who underwent both BSGI with [^99^mTc]Tc-Sestamibi and breast bioptic procedure.

The data of our study highlighted the possible use of BSGI with [^99^mTc]Tc-Sestamibi for the early identification of breast tumors with high risk to develop bone metastasis due to the presence of BOLCs. To characterize the presence of these cells, we studied the simultaneous expression of two well-known osteoblast biomarkers, RANKL and RUNX2 [25,26]. In detail, we demonstrated a significant association between [^99^mTc]Tc-Sestamibi uptake and both the number of RANKL- and RUNX2-positive breast cancer cells (BOLCs). Other biomarkers linked to bone metabolism have been evaluated in this study, as well as the association between their expression and [^99^mTc]Tc-Sestamibi uptake. Specifically, we evaluated the expression of ER, whose activation is related to the differentiation of osteoblasts [27,28]; BMP-2, which is the most important osteoblast inducing factor [29,30]; and PTX3 [31,32]. According to the association between BOLCs and [^99^mTc]Tc-Sestamibi uptake, we observed a similar trend for ER, BMP-2, and PTX3. Still, according to our preliminary investigation [20], [^99^mTc]Tc-Sestamibi uptake displayed a positive association with the cancer proliferative index, evaluated as a percentage of Ki67 positive cancer cells. As reported above, our experimental data may be explained by the physical–chemical properties of sestamibi as well as the presence and activities of the BOLCs in the breast cancers. Thus, here, we speculated that the increasing of sestamibi uptake in breast tumors was related to the presence of a high number of BOLCs actively involved in the production of microcalcifications mainly composed by HA or MgHAp. To further support this, we also investigated the possible association between the [^99^mTc]Tc-Sestamibi uptake and the occurrence of microcalcifications in breast lesions. Notably, in our patients, the uptake of ^99m^Tc sestamibi was higher in breast cancer lesions with MgHAp calcifications than in lesions with HA or without calcifications. No association was noted considering the presence of microcalcifications instead of their elemental composition.

The data that were reported here, if confirmed in a large cohort of patients, could impact the current management of bone metastasis from breast cancer. Indeed, the BSGI with [^99^mTc]Tc-Sestamibi analysis could represent a diagnostic tool capable to identify breast cancer patients with a high propensity to develop bone metastasis early. Stratification of patients according to the risk of bone metastasis development can be used to plan personalized therapeutic protocols and, in future, to develop new anti-BOLCs drugs. In this context, we recently hypothesized the use of anti-RANKL molecules (such as denosumab) in the treatment/prevention of bone metastasis from breast cancer [33]. The same molecules will be used as radiotracers capable to mark the BOLCs into the breast tumors.

## 5. Conclusions

In the era of personalized medicine, it has become clear that the biomedical translational research requires synergistic, transdisciplinary competencies. In this scenario, the departments of imaging diagnostic and pathology can build a scientific platform for the identification, development, and clinical applications of new biomarkers [34,35]. Following this approach, in this pilot study, we evaluated data from the nuclear medicine unit and the anatomic pathology department on breast cancer osteotropism to identify a new possible interpretation of BSGI with [^99^mTc]Tc-Sestamibi analysis.

## Figures and Tables

**Figure 1 jcm-09-00747-f001:**
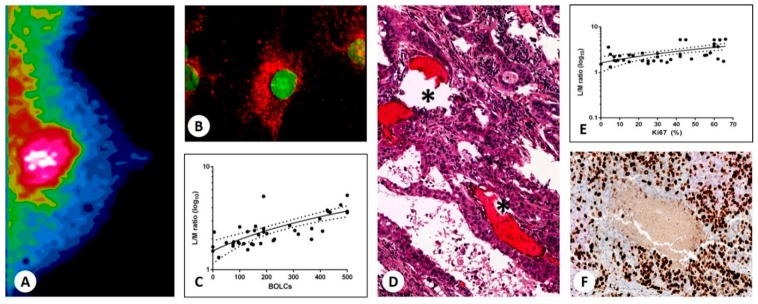
[^99^mTc]Tc-Sestamibi uptake in breast cancer. (**A**) Image shows [^99^mTc]Tc-Sestamibi uptake in a 55-year-old patient. (**B**) RUNX2- and RANKL-positive breast cancer cells. RANKL appears Texas-Red labeled, whereas RUNX2 is FITC-labeled. (**C**) Graph shows a significant association between lesion to non-lesion ratio and the number of breast osteoblast-like cells (BOLCs) (r^2^ = 0.5056; *p* < 0.0001). (**D**) Hematoxylin–eosin staining shows an infiltrating breast carcinoma with calcified structures (asterisks) similar to bone trabeculae. (**E**) Graph displays a significant association between the L/M ratio and the percentage of Ki67 positive cancer cells (r^2^ 0.3162; *p* = 0.0002). (**F**) Image shows several Ki67 positive breast cancer cells next to microcalcifications.

**Figure 2 jcm-09-00747-f002:**
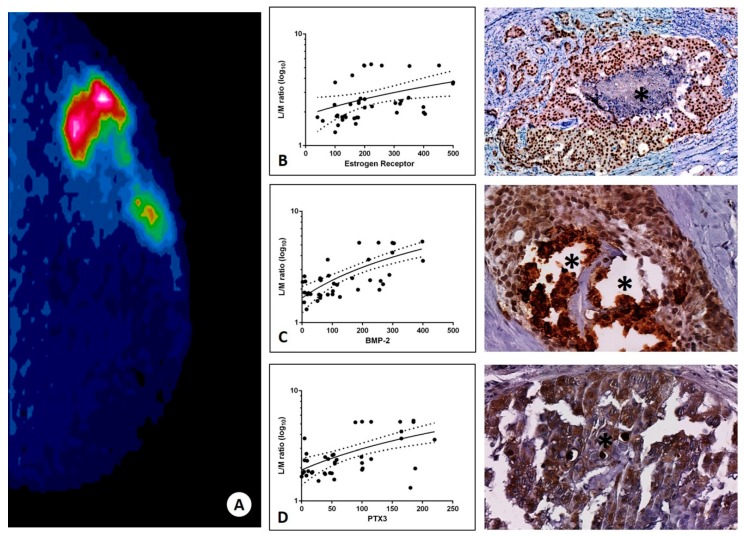
Association between [^99^mTc]Tc-Sestamibi uptake and the expression of bone biomarkers. (**A**) High [^99^mTc]Tc-Sestamibi uptake in a 53-year-old patient. (**B**) Graph shows a significant association between the estrogen receptor expression and the lesion to non-lesion ratio (r^2^ = 0.1527; *p* < 0.0208). Representative immunohistochemical image shows numerous estrogen positive breast cancer cells close to a microcalcification (asterisk). (**C**) Graph displays the positive association between sestamibi uptake and the number of BMP-2 positive tumor cells (r^2^ = 0.4953; *p* < 0.0001). Representative immunohistochemical image shows numerous estrogen positive breast cancer cells close to microcalcifications (asterisks). (**D**) Graph shows the significant association between the sestamibi uptake and the expression of PTX3 by cancer cells (r^2^ = 0.2975; *p* = 0.0003). High PTX3 expression of breast cancer cells next to a microcalcification (asterisk).

**Figure 3 jcm-09-00747-f003:**
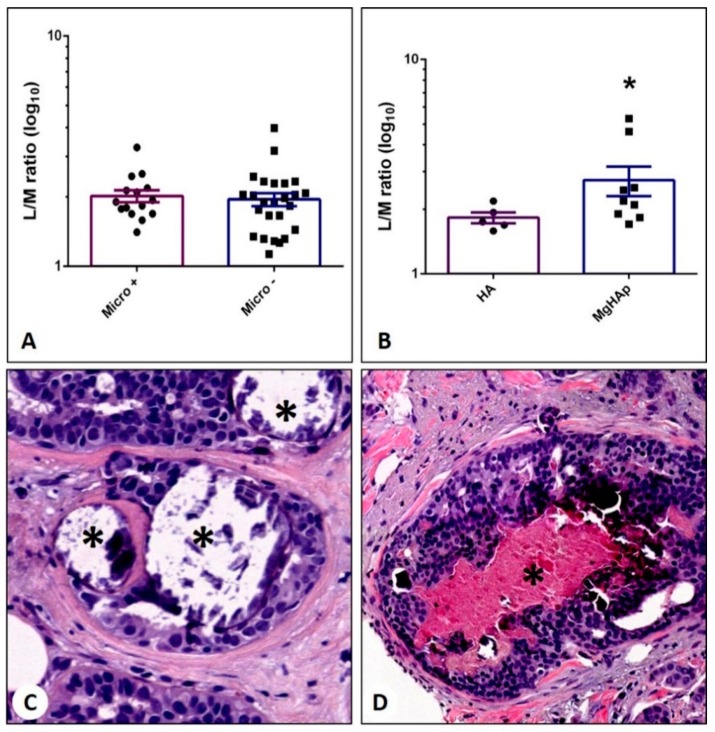
Association between [99mTc]Tc-Sestamibi and the presence of microcalcifications in breast cancer. (**A**) Graph shows no differences in terms of L/M ratio between breast cancers with (Micro+) and without (Micro-) microcalcifications. (**B**) Graph displays a significant increase in terms of lesion to non-lesion ratio in breast lesions with hydroxyapatite microcalcifications with respect to magnesium-substituted hydroxyapatite microcalcifications. (**C**) Hematoxylin–eosin staining shows microcalcifications (asterisks) made of hydroxyapatite in a breast infiltrating carcinoma. (**D**) Hematoxylin–eosin staining displays microcalcifications (asterisk) made of magnesium-substituted hydroxyapatite in a breast-infiltrating carcinoma.

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
