# Peer review of "Breast-Specific Gamma Imaging with [99mTc]Tc-Sestamibi: An In Vivo Analysis for Early Identification of Breast Cancer Lesions Expressing Bone Biomarkers"

_jcm, 2020, doi:10.3390/jcm9030747_

Round 1

Reviewer 1 Report

This manuscript presents the preliminary results of a pilot study. Investigation was to evaluate the possible relationship among 99mTc-sestamibi uptake, the presence of Breast Osteoblast-Like Cells and the expression of molecules involved in bone metabolism such as estrogen receptor, bone morphogenetic proteins-2 and PTX3. Authors of the study, based on the analysis of a first small number of patients, advance the hypothesis the [99mTc]Tc-Sestamibi may be considered a new tool for the management of patients affected by breast cancer. 

This pilot study is interesting but a few minor points need attention:

  1. Introduction.The radiopharmaceutical [99mTc]Tc-Sestamibi was predominantly utilised as a perfusion agent for cardiac imaging. Unexpectedly, it was also found to have accumulated in breast cancer tissues. It’s known and unexpected that the [99mTc]Tc-Sestamibiuptake and its distribution in breast cells has a value which is higher than that of myocardial cells whereas a minuscule amount of the administered activity is confiscated in the breast tissue.

It is requested to add in the introduction the rationale for the use of [99mTc]Tc-Sestamibi. This description is present in the discussion of the results, it should be reported in this section.

  1. Methods. In the various paragraphs (2.1, 2.2, 2.4) of this chapter the methods are indicated by means of a simple reference to published works. for easier reading, at least a schematic summary of the method used is recommended.

  1. Results. If available, the follow-up data of the patients analyzed would be interesting.

  1. Conclusion. Attention to conclusion in the “Abstract” are more complex than those in the conclusion paragraph. Given the small number of patients analyzed, I recommend greater caution in the conclusions.

Attention to the correct writing of the name of the radiopharmaceutical in the various points of the manuscript which must be the following: [99mTc]Tc-Sestamibi and not: 99TC-sestamibi or 99mTC sestamibi.

Attention to the use of acronyms, as at the first (or only) use they must be fully explained.

Author Response

Ref.: Manuscript: jcm-726594

"Breast-specific gamma imaging with 99mTc-sestamibi: an in vivo analysis for early identification of breast cancer lesions expressing bone biomarkers"

Submitted to: Journal of Clinical Medicine (Section Nuclear Medicine & Radiology - Special Issue Emerging Technologies for Medical Imaging - Diagnostics, Monitoring and Therapy of Cancers)

Before we begin the point by point review of the list of concerns, we would like to thank the Reviewer for their comments on how to improve the manuscript, which has been revised accordingly, as well as the Editors for calling for a new submission of an improved version of our manuscript.

Reply to Reviewer 1

This manuscript presents the preliminary results of a pilot study. Investigation was to evaluate the possible relationship among 99mTc-sestamibi uptake, the presence of Breast Osteoblast-Like Cells and the expression of molecules involved in bone metabolism such as estrogen receptor, bone morphogenetic proteins-2 and PTX3. Authors of the study, based on the analysis of a first small number of patients, advance the hypothesis the [99mTc]Tc-Sestamibi may be considered a new tool for the management of patients affected by breast cancer.

Reply: we would like to thank the Reviewer for expressing interest in our work, and for their availability to review a revised version of our manuscript.

This pilot study is interesting but a few minor points need attention:

 Introduction.The radiopharmaceutical [99mTc]Tc-Sestamibi was predominantly utilised as a perfusion agent for cardiac imaging. Unexpectedly, it was also found to have accumulated in breast cancer tissues. It’s known and unexpected that the [99mTc]Tc-Sestamibiuptake and its distribution in breast cells has a value which is higher than that of myocardial cells whereas a minuscule amount of the administered activity is confiscated in the breast tissue.

It is requested to add in the introduction the rationale for the use of [99mTc]Tc-Sestamibi. This description is present in the discussion of the results, it should be reported in this section.

Reply: Thank you for highlighting this possible source of confusion in the previous version of the paper. In the revised form of our manuscript we added the following consideration in the “Introduction” paragraph.

Recently, the American College of Radiology reported detailed indications for the use of dedicated breast gamma imaging, Breast Specific Gamma Imaging – (BSGI) and Molecular breast imaging (MBI), in order to studied the extent of disease/preoperative staging in de novo detected breast tumors, assessment of response to the treatment with neoadjuvant chemotherapy, identification of local recurrence, assessment for primary breast lesions in patients with unidentified primary, screening for high risk women who cannot undergo MRI and MBI as adjunct to conventional breast imaging for problem solving in unclassified cases

Methods. In the various paragraphs (2.1, 2.2, 2.4) of this chapter the methods are indicated by means of a simple reference to published works. for easier reading, at least a schematic summary of the method used is recommended.

Reply: Thank you for this pointing out.  We added more informations in the Methods section.

Results. If available, the follow-up data of the patients analyzed would be interesting.

Reply: Thank you for this pointing out.  Unfortunately, no follow-up informations are available for this cohort of patients. We are performing a similar study on a cohort of breast cancer patients with long-time follow-up.

Conclusion. Attention to conclusion in the “Abstract” are more complex than those in the conclusion paragraph. Given the small number of patients analyzed, I recommend greater caution in the conclusions.

Reply: Thank you for this pointing out.  We modified the abstract in agreement with the reviewer’ indications.

Attention to the correct writing of the name of the radiopharmaceutical in the various points of the manuscript which must be the following: [99mTc]Tc-Sestamibi and not: 99TC-sestamibi or 99mTC sestamibi.

Attention to the use of acronyms, as at the first (or only) use they must be fully explained.

Reply: Done

Reviewer 2 Report

Manuscript title: Breast-Specific Gamma Imaging with 99mTc-Sestamibi: An In Vivo Analysis for Early Identification of Breast Cancer Lesions Expressing Bone Biomarkers by Nicoletta Urbano et al.

This clinical paper describes the role of 99mTsestamibi imaging in the possible relationship with the presence of breast osteoblast-like cells (BOLCs), expression of various receptors and metastasis into bone tissue. The authors found that the identification and increased presence of 99mTc-sestamini positive cells may increase the metastatic potential. This is an important finding.

To increase the potency of this study it is wishful to describe the imaging procedure and of taking biopts into more detail. Is this done by laparoscopy and have all (when more lesions were present) the cancerous material been removed and evaluated as well? Is the SPECT scan used as image guided resection and after taking the biopts have the tissues been checked for remaining  cancerous material using the radioactivity signal?

To understand the role of sestamibi uptake and the relation with metastasis in this respect the mechanisms of tracer uptake needs to be discussed. At least, what will be the patient management depending on the outcome of this research?

After finding 99mTc-sestamibi positive material, what are the next steps to take for the patient and what are the difference compared to patients without 99mTc-setstamibi positive cells, but still with suspicion for having breast cancer? Is the recidivism known between these 3 groups? Imaging breast cancer is found to have 77% of specificity and there are some concerns related to false positives. How can this be of influence on the outcome of the results in this study?

The description of the Immunohistochemistry in paragraph 2.3 has not been entirely described into detail. In this way it is hard to reproduce the results.

On page 6 in the Discussion section, lines 181-184, it is stated that cationic nature of sestamibi is responsible for accumulation into the mitochondrion. However, it is known that tumor cells expose anionic groups on the outer membrane, which is more prominent in fast replicating cells. One should expect accumulation of the tracers based on this characteristic. How does this fit in the proposed uptake mechanism in the Discussion section?

Minor comments:

Line 118: Clarify L/N ratio.

Line 128: Clarify L/M value.

Legend to Figure 1,2 and 3: contain various typo’s.

Figure 2A: indicate with an arrow the tumor.

Graphs in Figure 1 and 2 are without a significant p-value. Is correlation calculated or known?

The grammar need to be verified. E.g. sentences at 81-83, 90-91.

Be consistent in using superscript 99mTc in the paper (lines 20, 23, 68, 178, 192, 195).

Author Response

Ref.: Manuscript: jcm-726594

"Breast-specific gamma imaging with 99mTc-sestamibi: an in vivo analysis for early identification of breast cancer lesions expressing bone biomarkers"

Submitted to: Journal of Clinical Medicine (Section Nuclear Medicine & Radiology - Special Issue Emerging Technologies for Medical Imaging - Diagnostics, Monitoring and Therapy of Cancers)

Before we begin the point by point review of the list of concerns, we would like to thank the Reviewer for their comments on how to improve the manuscript, which has been revised accordingly, as well as the Editors for calling for a new submission of an improved version of our manuscript.

Reply to Reviewer 2

Manuscript title: Breast-Specific Gamma Imaging with 99mTc-Sestamibi: An In Vivo Analysis for Early Identification of Breast Cancer Lesions Expressing Bone Biomarkers by Nicoletta Urbano et al.

This clinical paper describes the role of 99mTsestamibi imaging in the possible relationship with the presence of breast osteoblast-like cells (BOLCs), expression of various receptors and metastasis into bone tissue. The authors found that the identification and increased presence of 99mTc-sestamini positive cells may increase the metastatic potential. This is an important finding.

Reply: we would like to thank the Reviewer for expressing interest in our work, and for their availability to review a revised version of our manuscript.

To increase the potency of this study it is wishful to describe the imaging procedure and of taking biopts into more detail. Is this done by laparoscopy and have all (when more lesions were present) the cancerous material been removed and evaluated as well? Is the SPECT scan used as image guided resection and after taking the biopts have the tissues been checked for remaining  cancerous material using the radioactivity signal?

Reply: Thank you for this pointing out.  In the paragraph “2.1. 99Tc-Sestamibi-High Resolution SPECT” we added more detailed information according reviewer ‘suggestion (see pag. 3).

To understand the role of sestamibi uptake and the relation with metastasis in this respect the mechanisms of tracer uptake needs to be discussed. At least, what will be the patient management depending on the outcome of this research?

Reply: Thank you for this pointing out.  In the discussion paragraph we added the consideration suggested by reviewer.

Specifically, the text was modified as follow:

Data here reported, if confirmed in a large cohort of patients, could impact the currently management of bone metastasis from breast cancer. Indeed, the BSGI with [99mTc]Tc-Sestamibi analysis could represents a diagnostic tool capable to early identify breast cancer patients with high propensity to develop bone metastasis. Stratification of patients according to the risk of bone metastasis development can be used to plan personalized therapeutic protocols and, in future, to develop new anti-BOLCs drugs. In this context, we recently hypothesis the use of anti-RANKL molecules (such as denosumab) in the treatment/prevention of bone metastasis from breast cancer [33]. The same molecules will be used as radiotracers capable to mark the BOLCs into the breast tumors.

After finding 99mTc-sestamibi positive material, what are the next steps to take for the patient and what are the difference compared to patients without 99mTc-setstamibi positive cells, but still with suspicion for having breast cancer? Is the recidivism known between these 3 groups? Imaging breast cancer is found to have 77% of specificity and there are some concerns related to false positives. How can this be of influence on the outcome of the results in this study?

Reply: we would like to thank the Reviewer for these questions related to our manuscript. Unfortunately, for this case selection we have neither follow-up nor recidivism data. We performing a similar study on a cohort of breast cancer patients with long-time follow-up. Considering this as a pilot study, is it difficult introduce the considerations suggest by reviewer. Of course, we will use these indications in our future studies.

The description of the Immunohistochemistry in paragraph 2.3 has not been entirely described into detail. In this way it is hard to reproduce the results.

Reply: done

On page 6 in the Discussion section, lines 181-184, it is stated that cationic nature of sestamibi is responsible for accumulation into the mitochondrion. However, it is known that tumor cells expose anionic groups on the outer membrane, which is more prominent in fast replicating cells. One should expect accumulation of the tracers based on this characteristic. How does this fit in the proposed uptake mechanism in the Discussion section?

Reply: Thank you for pointing this out. We modified the discussion paragraph accordingly reporting more information about the kinetic of sestamibi.

Minor comments:

Line 118: Clarify L/N ratio.

Line 128: Clarify L/M value.

Reply: Done

Legend to Figure 1,2 and 3: contain various typo’s.

Reply: Done

Figure 2A: indicate with an arrow the tumor.

Reply: we modified the image and added an arrow to mark the tumor.

Graphs in Figure 1 and 2 are without a significant p-value. Is correlation calculated or known?

Reply: The p values were reported in the main text. We added these also in the figure legend.

The grammar need to be verified. E.g. sentences at 81-83, 90-91.

Reply: Done

Be consistent in using superscript 99mTc in the paper (lines 20, 23, 68, 178, 192, 195).

Reply: Done

This manuscript is a resubmission of an earlier submission. The following is a list of the peer review reports and author responses from that submission.